# Barriers and facilitators to digital technology application for antimicrobial resistance surveillance: A co-produced qualitative synthesis

**Ayodele Akinyele**[1‡], **Emmanuel Olamijuwon**[2‡], **Abeeb A. Adeniyi**[1],
**Oluwatobiloba S. Kazeem**[1,3], **Michael Popoola**[4], **Tochukwu C. Agboeze**[1],
**Iruka N. Okeke**[1,3]*

1 Department of Pharmaceutical Microbiology, Global Health Research Unit for Genomic Surveillance of Antimicrobial Resistance, Faculty of Pharmacy, University of Ibadan, Ibadan, Oyo State, Nigeria, 2 School of Geography and Sustainable Development, University of St Andrews, Scotland, United Kingdom, 3 Department of Medical Microbiology and Parasitology, College of Medicine, University of Ibadan, Ibadan, Oyo State, Nigeria, 4 Nigeria Centre for Disease and Control, NCDC, Abuja, Nigeria

‡ These authors are joint senior authors on this work.
* iruka.n.okeke@gmail.com

## Abstract

The systematic collection, analysis, and interpretation of antimicrobial resistance (AMR) data are imperative to quantify the AMR burden, monitor and identify emerging AMR, and inform global, international, and national health strategies and guidelines. Despite ongoing global efforts to improve surveillance capacities across Nigeria and other African countries, laboratory information management systems (LIMS) that could improve data quality, and completeness remain underutilized. We used a participatory research approach, drawing on the unique experiences of various stakeholders, such as data analysts, laboratory scientists, infection prevention and control specialists, medical doctors, and representatives from the National Coordinating Center in Nigeria. Over two phases of evidence synthesis, involving in-depth interviews and a participatory co-design workshop, we sought to understand the experiences of key stakeholders in using the LIMS tool, WHONET, for AMR surveillance, and co-develop solutions and priorities to address the challenges they experience. We identified a complex interplay of systemic/political factors and structural/user-related factors that influence the use of WHONET as a LIMS. Key areas for intervention identified by stakeholders include addressing infrastructural deficits, enhancing stakeholder engagement, and improving the perceived usefulness of the system, as well as the need for management support. Stakeholders also identified 18 potential solutions to tackle key challenges, ten of which require low effort and have a high influence on LIMS use behaviors. Our study highlights the multifaceted challenges affecting the effective utilization of WHONET for AMR surveillance in Nigeria. The co-developed solutions provide a roadmap for targeted interventions to strengthen AMR surveillance capacity and inform evidence-based public health strategies.

**Data availability statement:** All data used for the study are included in the paper.

**Funding:** This project was funded by the National Institute of Health Research Global Health Research Unit on Genomics and enabling data for Surveillance of Antimicrobial Resistance Consortium (Award #NIHR133307 to INO). INO is a Calestous Juma Science Leadership Fellow supported by the Bill and Melinda Gates Foundation (INV-036234). AA is funded by the Robert Koch Institute's CARe Award (ZV 2.2.6 / 1368-2057). EOO is funded by the Wellcome Trust Discretionary Award (228281/Z/23/Z). The funders had no role in this paper's content, crafting, or submission.

**Competing interests:** The authors have declared that no competing interests exist.

## 1. Introduction

Antimicrobial resistance (AMR) is a global health crisis that endangers overall well-being. Its impacts are far-reaching, affecting not just human health but also animal health and natural ecosystems. The mortality and morbidity rates associated with AMR are concerning and are projected to increase if decisive action is not taken, particularly impacting low and middle-income countries [1]. Close to five million deaths were estimated to be associated with bacterial AMR, including 1.14 million deaths attributable to bacterial AMR in 2019. Modelling data suggest that western sub-Saharan Africa bears the highest global burden, with 27.3 deaths per 100,000 in 2019 [2]. Although these estimates are derived from limited data and likely under-estimate the AMR burden, the huge burden of AMR underscores an urgent need for effective interventions and collaborative efforts spanning several sectors, particularly those related to health system surveillance.

The systematic collection, analysis, and interpretation of data on antimicrobial resistance (AMR) is imperative to quantify the burden, monitor and identify emerging AMR, and inform global, international, and national health guidelines [3,4]. Comprehensive and complete data on pathogen distribution, resistance prevalence, and infection-related mortality are valuable resources for estimating AMR burden and can potentially optimize health system functioning and enhance emergency preparedness [5,6]. Multiple AMR and antimicrobial consumption surveillance platforms, such as the Global Antimicrobial Resistance and Use Surveillance System (GLASS), among others, currently operate worldwide to strengthen the AMR evidence base and inform policies for AMR control [1,7]. The success of these surveillance platforms and the capacity of countries to contribute data to them depend on several factors, including the health system strength and connectivity, laboratory operations and quality, as well as diagnostic and data handling capabilities [8]. Data generated from newer AMR surveillance networks in LMICs is often characterized by several shortcomings, such as unreported or under-reported clinical, phenotypic antimicrobial susceptibility testing (AST), and other crucial data parameters, and the lack of standardized laboratory practices [6,9,10]. In a recent assessment of bacteriology laboratories across 14 African countries, less than one-fifth of the laboratories managed data electronically [8]. As a result, there is a global call for the integration of technology systems in laboratory information management to enable a comprehensive understanding of the magnitude of AMR and to direct policy action.

WHONET, a free desktop application for managing and analyzing bacteriological culture and susceptibility testing data, offers numerous opportunities to gather, curate, interpret, and submit data to GLASS [11]. The application is being used in more than 120 countries [12]. Besides being designed to facilitate the systematic collection of pathogen and patient data, as well as phenotypic AMR surveillance, its SaTScan package facilitates the detection of suspected outbreaks, enabling timely infection prevention and control (IPC) interventions in hospital environments [13,14]. Additionally, WHONET detects data entry errors and ensures consistency in data collection by standardizing the types of data collected and the variable names used across laboratories, ultimately improving data accuracy and reliability. This data tool

is useful at the facility level and provides insights regionally, nationally, and globally when aggregated with data from other facilities. However, despite its enormous potential, WHONET software is underutilized at many sentinel sites, leading to incomplete metadata associated with resistance. As global efforts to improve laboratory and surveillance capacities expand across LMICs, more research is needed to understand the barriers and facilitators of the efficient use of information technologies, such as laboratory information management systems for AMR surveillance [9,12].

To date, an extensive body of research has explored the adoption and use of new technologies across various fields, including education, tourism, and global health. These studies, grounded in diverse theoretical frameworks [15–19], have shown that perceived usefulness, ease of use, social influence, and motivation are key determinants of new technology use behavior. However, the adoption and use of laboratory information management systems for AMR surveillance present a uniquely complex challenge. These systems operate within a multifaceted and dynamic environment, often involving multiple stakeholders whose roles are deeply interconnected and interdependent. The actions and decisions of one stakeholder can also generate ripple effects, influencing the behavior and outcomes of others within the system. This complexity necessitates a more nuanced understanding of the behavioral, organizational, technical, and systemic factors that shape the use of laboratory information management systems for antimicrobial resistance surveillance. Despite this, a clear picture of the barriers and facilitators of efficient use of laboratory information management tools and systems (LIMS) in microbiology laboratories has yet to emerge.

In this study, we sought to understand the barriers and facilitators of efficient use of laboratory information management tools and systems (LIMS), particularly WHONET, in microbiology laboratories among key stakeholders engaged in AMR surveillance in Nigeria. We focus on Nigeria, an LMIC in sub-Saharan Africa, where national surveillance formally debuted in 2017 when the country registered with GLASS. Nigeria's surveillance system is operated by the Nigeria Centre for Disease Control and Prevention (NCDC), with support from the University College Hospital Ibadan and the Global Health Research Unit for Genomic Surveillance of Antimicrobial Resistance at the University of Ibadan, which together provide reference laboratory services [20]. Between 2017 and 2023, the surveillance system expanded from three to eleven sentinels. As recommended by WHO GLASS, Nigeria's National Coordinating Center at NCDC requires the use of WHONET to gather, curate, and submit data to GLASS [11]. However, like many other LMICs, the adoption and efficient use of WHONET at sentinel labs remains limited. Thus, by examining the perspectives, challenges, and experiences of stakeholders, this research aims to provide a comprehensive understanding of the factors influencing the adoption of laboratory information systems for AMR surveillance and to identify opportunities for addressing their issues in an LMIC. Consequently, findings from this study can inform the development of context-specific acceptable interventions to motivate the effective use of laboratory information systems as global efforts to improve AMR surveillance expand across Africa and other limited-resourced settings.

## 2. Methods

### 2.1. Ethics statement

Ethical approval for this study was obtained from the UI/UCH Ethics Review Committee board with IRB number UI/EC/24/0512. All participants gave written informed consent before participating in the key informant interviews and participatory workshops.

### 2.2. Study design

This research was conducted at the sentinel sites across Nigeria highlighted in Fig 1. Since 2019, the Global Health Research Unit for Genomic Surveillance of Antimicrobial Resistance (GHRU-GSA) has worked with the NCDC to strengthen AMR surveillance and data management in Nigeria. Nigeria is also a recipient of a Fleming Fund country grant, which strengthened capacities to generate antimicrobial susceptibility testing data across 12 sentinel sites and NCDC reference laboratories. Sentinel laboratories are expected to routinely enter and manage their data in WHONET, software

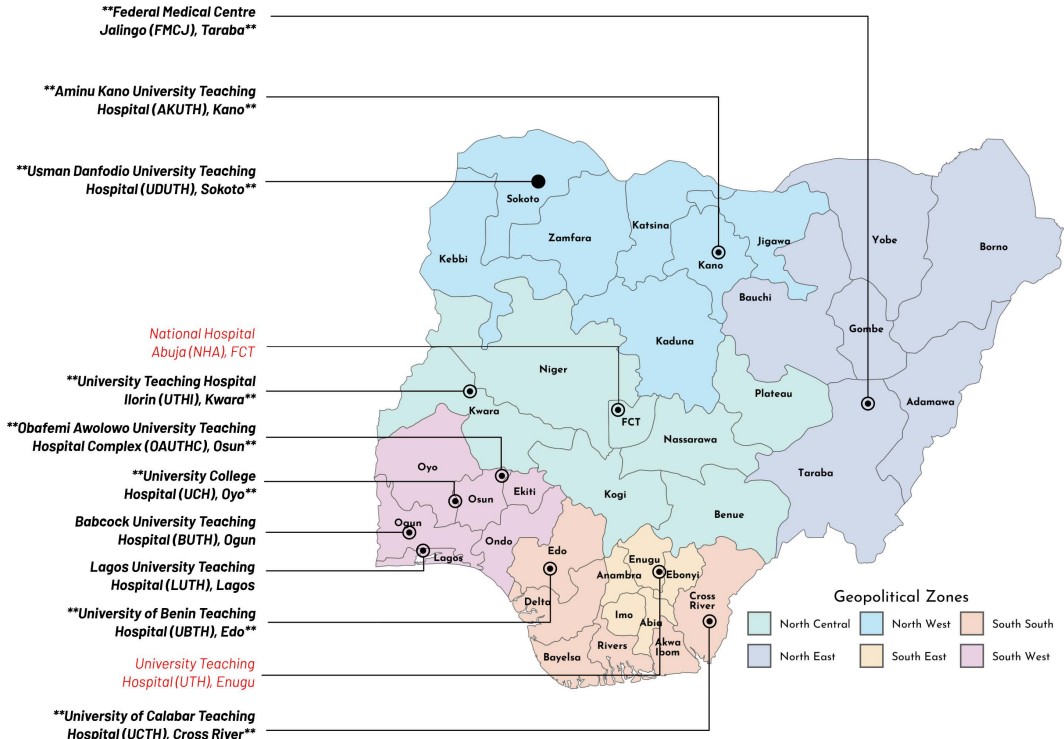

**Fig 1. Nigerian map showing the geo-political location of sentinels currently participating in the national AMR surveillance programme** *((Note: Base layer map for Nigeria was obtained from Natural Earth.* https://www.naturalearthdata.com/downloads/*; Bolded black labels with asterisks indicate sentinel sites where participants were recruited for all phases of the study, bolded black labels without asterisks indicate sentinel sites where participants were recruited for the co-design workshop and feedback session, and red labels indicate sentinel sites that did not participate in this study).*

developed and supplied by WHO for managing AST data, particularly in resource-limited settings [11]. In 2022, GHRU launched actions to optimize data flow within the surveillance network to improve network functions and use these insights to guide evidence-based policy-making.

## 2.3. Data collection

We adopted a participatory action research (PAR) approach with key stakeholders at sentinel laboratories across the country. This research method is oriented towards community action and analysis to address complex social problems, including those related to data use for public health policy and programming [21,22]. Accordingly, the PAR methodology has the potential not only to generate people-centered solutions but also to motivate behavioral change among stakeholders. It offers stakeholders the opportunity to be involved in multiple recurring stages (phases) of community-based observation, reflection, planning, and action, with each phase building upon and influencing subsequent phases. To our knowledge, this work is the first to develop participatory decision-making approaches to enhance the use of laboratory information management systems for antimicrobial resistance surveillance, particularly in limited-resource settings.

We conducted multiple phases of PAR, including an initial engagement, in-depth interviews, a collaborative workshop, and a dissemination and feedback meeting with stakeholders from all sentinel labs, to capture the breadth of drivers and potential solutions that motivate behavioral change. Only ten of the 12 participating sentinel labs in the country consented to participate in the study (Fig 1). At each of the ten sentinel sites, we held an initial stakeholder meeting between August 27 and October 5, 2024, where we introduced the present study and invited key stakeholders to co-design the process by

nominating key contacts for further engagement. We targeted key personnel involved in data capture, as well as their line managers and infection prevention and control officers, who are stakeholders in the use of data. Subsequently, we scheduled appointments to conduct an in-depth, in-person interview with the nominated stakeholder (Table 1). Eight of the initial ten sentinels were available to participate in this in-depth interview phase. Four out of the eight sentinel laboratories have at one or more full-time personnel responsible for data entry and report submission to the National Coordinating Center, while the remaining four sentinel laboratories do not. The interviews focused primarily on gaining in-depth understanding of the stakeholders' experiences with using the laboratory information management system in microbiology laboratories for AMR surveillance. A detailed list of questions asked during the interviews is available in S1 Text. The data derived from the in-depth interview provided critical and in-depth insights into the barriers and facilitators of sub-optimal use of the laboratory information management system for AMR surveillance. The interviews were conducted in English and facilitated by a field interviewer, supervised by AyA and AbA, using an interview guide. Both EO and AyA have formal training in qualitative research methodologies and have trained all assistants prior to the fieldwork. The interviews were tape-recorded, and notes were taken to ensure that no information was omitted during the discussion/interaction. Most of the interviews lasted for 45 minutes or less.

Following the initial stakeholder meeting and in-depth interviews, we held a participatory co-design workshop on October 6, 2024, which was structured to systematize subjective perspectives into shared forms of knowledge on the nature of the problem and build consensus on actions to address the identified issues. The workshop was organized as part of an already planned data workshop in October 2024. Ten of the twelve participating sentinels were available to participate in the workshop. During this workshop, participants were encouraged to reflect on the problems identified in the previous phase and co-develop contextually relevant strategies to address the challenges that had been identified. Given the diversity in the stakeholder profile and potential conflict of interest, we stratified the participants into two groups—one with line managers and IPC officers, and another group involving data capture officers and laboratory scientists. Discussions in the groups were facilitated and recorded on prepared flip charts to display a collective record (AyA, AbA, and EO), allowing for checking and rechecking of consensus views. A detailed list of the key issues discussed during this phase is available in S2 Text. The workshops in both groups were held in English, and workshop guides were used to structure the discussions. With separate permissions, workshops were audio-recorded and transcribed verbatim.

In the final phase, we consolidated findings from the previous phases and presented them to all the participants on October 8, 2024. Subsequently, the group reconvened and shared their feedback. We then triangulated information from this phase with the previous phase (in-depth interviews) to co-develop a holistic framework for understanding critical challenges to the effective use of the laboratory information management system/tool.

## 2.4. Data analysis

Audio recordings from the in-depth interviews were transcribed to aid further analysis. The transcripts were then imported and analyzed in NVIVO (v.8). We leveraged Braun and Clarke's [23–25] reflexive method for thematic analysis to code and analyze the data. This analytical approach is widely used in qualitative research and is a useful method for

**Table 1. Composition of discussion groups.**

|  | Data officer | Facility IPCP | National Coordinating Center personnel | Medical technologists | Medical doctors/ Consultants | Laboratory technologist/scientist | Total |
|---|---|---|---|---|---|---|---|
| In-depth-interview | 4 | 0 | 0 | 0 | 0 | 19 | 23 |
| Participatory co-design workshop | 5 | 2 | 0 | 2 | 3 | 15 | 27 |
| Feedback session | 5 | 2 | 3 | 2 | 3 | 15 | 30 |

Note: IPCP: Infection prevention and control personnel

summarizing the key features of large datasets, examining the perspectives of different research participants, highlighting emergent themes, commonalities, and differences, and generating unanticipated insights to produce a clear and comprehensive analysis of the problem under study [23].

Analysis of the transcribed data began with reading and re-reading (EO) the texts transcribed from the in-depth interviews to become deeply familiar with them. We also developed reflective thoughts and ideas about coding during this analytical phase. We combined deductive thematic coding informed by the technology acceptance [26,27] and inductive coding that allowed themes to emerge from the data. Thematic codes were developed and organized until no new themes emerged. Succinct labels that capture the participants' views and experiences were subsequently created. The coded data extracts for each candidate theme were reviewed accordingly to ensure an articulate discourse pattern.

We utilized systems thinking tools [28,29], specifically causal loop diagrams (CLDs), to visualize and describe the interrelationships between issues identified during the in-depth interviews and the participatory co-design workshop. An initial CLD was developed (EO) using the Visual Paradigm online tool, with extracted codes from transcribed in-depth interviews. This initial CLD was then revised based on recommendations from participants who attended the participatory co-design workshop. The revised CLD, therefore, reflects the shared experiences of the various stakeholders and interrelationships across different issues or codes. We also developed a Cartesian graph to visualize and map proposed solutions identified by stakeholders based on the level of effort required and the potential influence of the intervention. We classified interventions as *high effort* if they require contributions from stakeholders across multiple levels (such as data analysts, hospital management, National Coordinating Center, and others) and *low effort* if only the intervention of stakeholders at one level is required. Interventions were also classified as *high influence* if their impact was sufficient to motivate behavior change in the absence of other interventions, and as *low influence* if their impact was insufficient to motivate behavior change in the absence of other interventions.

Validation of the CLDs, Cartesian graph, and the underlying conceptual thinking was carried out in the last phase, attended by all stakeholders. The objective was to determine whether they found the CLDs to be reasonable and reflect their views and experiences. All stakeholders present at the dissemination workshop were asked to generally assess the CLD and Cartesian graph. Consideration was given to how each code reflects shared experiences and the extent to which proposed solutions are contextually relevant. After an initial explanation of the key concepts (EO), all the respondents became familiar and understood the CLD for the first time. Participants were also asked to reflect on each link, indicating whether they agreed with it and whether any links were missing. The validation process contributed to further modification, leading to the final CLDs and the Cartesian graph presented in this paper.

### 2.5. Patient and public involvement

It was neither necessary, appropriate, nor possible to involve patients or the public in the design, conduct, reporting, or dissemination plans of our research, as the study focused on technology use within hospital settings for antimicrobial resistance surveillance. Nevertheless, we involved administrators, staff, and management of the facilities where this study was conducted in the conduct, analysis, reporting, and dissemination of the research findings.

## 3. Results

### 3.1. Participant characteristics

Table 1 summarizes the distribution of participants across the various phases of the workshop. Most of the participants across all phases of the study were laboratory technologists/scientists. The participatory workshop also included most of the relevant stakeholders except the National Coordinating Center. Table 2 summarizes the sociodemographic characteristics of participants engaged in the in-depth interviews. Most of the participants (52%) were aged 30–39 years, and about 83% were male. The in-depth interview also mainly comprised (83%) laboratory technologists/scientists with various

Global Public Health
PLOS

**Table 2. Socio-demographics of KII participants.**

| Variables | Participants (n = 23) |
|---|---|
| **Age** | |
| 30–39 | 12 |
| 40–49 | 8 |
| 50–59 | 3 |
| **Mean age** | 40.48 ± 7.12 |
| **Gender** | |
| Male | 19 |
| Female | 4 |
| **Specialty** | |
| Laboratory Technologist/Scientist | 19 |
| *Biomedical Science* | *(16)* |
| *Science Laboratory Technology* | *(1)* |
| *Genetics/Biotechnology* | *(1)* |
| *Microbiology* | *(1)* |
| Data scientists (or entry officer) | 4 |
| *Computer Science* | *(2)* |
| *Pure and Applied Physics* | *(1)* |
| *Statistics* | *(1)* |
| **Years of experience capturing data** | |
| 0–5 | 10 |
| 6–10 | 9 |
| 10+ | 4 |
| **Level of expertise with capturing data** | |
| Beginner | 1 |
| Intermediate | 13 |
| Expert | 9 |
| **Commitment level** | |
| Highly Committed | 9 |
| Committed | 6 |
| Not committed until assigned | 1 |
| Willing to volunteer | 7 |

academic backgrounds (such as biomedical science—16, microbiology—1, among others) and data scientists (or entry officers, 17%). Most of the interviewees (57%) were mid-career and had been capturing data on the bench for at least six years. Only one interviewee rated their level of expertise with using the LIMS as a beginner.

### 3.2. Barriers and challenges to effective use of laboratory information systems

Analysis of the transcribed qualitative data from the in-depth interviews and participatory co-design workshop revealed multiple systemic/political and structural/user behavioral factors that shape the effective use of WHONET, the nationally recommended laboratory information management system/tool (Fig 2).

**3.2.1. Structural/user behavioral factors.** Most of the participants affirmed the importance of laboratory information management systems for AMR surveillance, particularly because *"manual processes are outdated and delay reporting; digital systems are far more efficient"*. Participants also remarked on the utility of WHONET and its ease of use. One

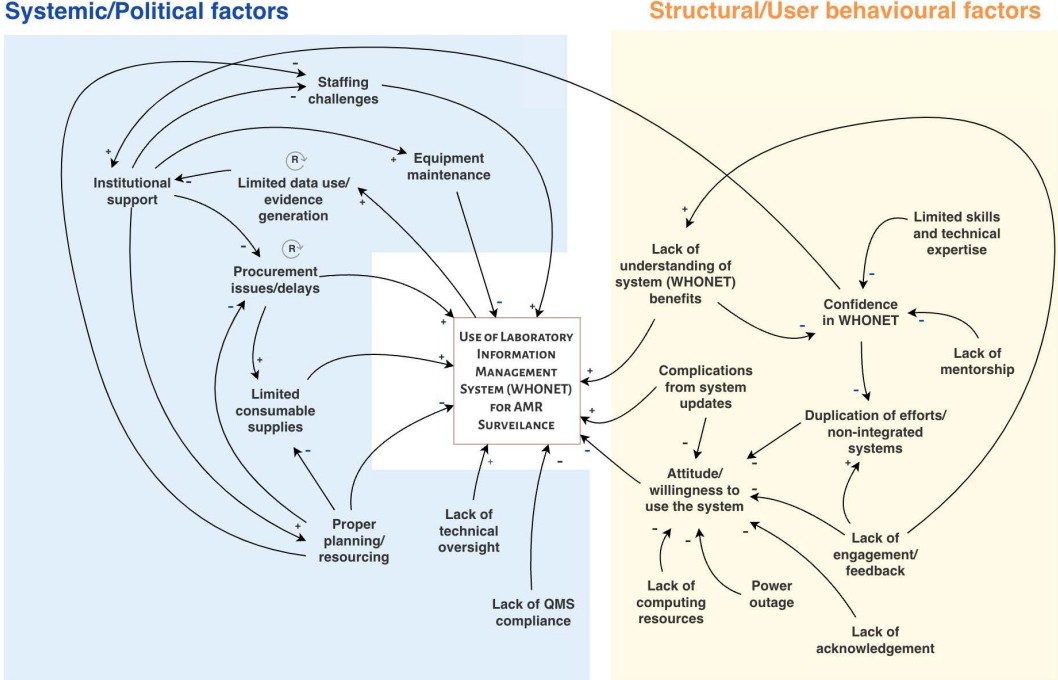

**Systemic/Political factors**

**Structural/User behavioural factors**

**Fig 2. Causal loop diagram showing inter-relationships among the barriers and facilitators of WHONET use for AMR surveillance** *(Note: Signs [-/+] beside the arrows depict the direction of the association [positive/negative]; QMS: Quality management system).*

participant acknowledged the ability of WHONET to reduce the likelihood of making mistakes through the built-in error checker. Despite the technology's user-friendliness, some participants acknowledged that complications arising from frequent system updates could discourage sustained use of the system, and further training could bridge this challenge. As one participant *(KII_B4)* notes: *"The system is straightforward and easy to use. However, continuous training is necessary, especially as system updates and new functions are introduced."* The participants' limited technological skills also shape how they resolve complications arising from system updates, which impacts their attitude towards using the system.

Infrastructural and technical barriers also emerged as key factors in the use of laboratory information management systems. For example, *"frequent power cuts make it hard to keep data updated, and backups are essential to avoid data loss--(KII_A1)".* This is especially true in these settings where WHONET is installed on very old desktop computers with outdated hardware and insufficient battery power, partly because one of WHONET's stated advantages for low-resource settings is that it can run on outdated hardware. The unstable supply of electricity and lack of computing facilities with adequate battery capacity led users to consider alternative record-keeping systems, such as capturing in notebooks as a backup, even though it is *"exhausting and time-consuming",* as one participant notes. This issue, however, is not in isolation from others. In addition to capturing data in notebooks as a backup (all eight sentinel labs), several participants from various laboratories reported that they were also required to use multiple LIMS, which were not interoperable. For instance, all federal hospitals in Nigeria are mandated to use the Health-in-a-Box system, the default hospital management information system, to capture patient demographics, geographical location, and billing details. However, this system does not capture laboratory-specific information, such as AST results, isolate identifications (IDs), or biochemical data, in a retrievable and analyzable manner, as WHONET does. In addition, these sentinel labs often double-enter data on the primary LIMS (WHONET) alongside other LIMS used at their facilities, such as SEDRILIMS (OAUTHC), RAPID

MEDCHART (UBTH), Health-in-a-Box (AKTH, UCH and UCTH), and REDCap (UDUTH), daily or occasionally for further data capturing. Five (5) out of the eight (8) sentinel labs (AKTH, FMCJ, UCTH, UDUTH, and UITH) also use Microsoft Excel either regularly/or for exporting data to WHONET. As a result, laboratories manually enter the same data (patient information alongside laboratory-specific data) into multiple systems. As reported by one participant, this duplication of efforts *"adds to our workload"*, particularly in a health system that is already overburdened, *"increases the chances of errors",* and decreases motivation and attitude towards the use of LIMS. Although WHONET can be made interoperable using BacLink, this is a specialist function that is only used by two of the eight sentinel labs (UCTH and UCH). In contrast, others reported non-use due to limited technical capacity, compatibility issues with their existing systems, or because they do not have any data backlog to import into WHONET.

Additionally, participants highlighted the lack of recognition in reports, publications, and informally as a key influence on their moral and long-term commitment to inputting laboratory results and relevant metadata into the LIMS. This was particularly important, given that many of them reported working voluntarily on the bench to capture the data. There were multiple references to senior management officers using *"our data"* for *"their research",* demonstrating an unshared sense of belonging in the creation of a national AMR surveillance system. The lack of acknowledgement and involvement also shapes attitude and motivation to effectively input laboratory results and relevant metadata efficiently.

**3.2.2. System/political factors.** Stakeholders also identified multiple systemic and political barriers and challenges to the effective use of laboratory information management systems. A recurring theme identified during the in-depth interviews and co-design workshop is the lack of institutional support, which also impedes recruitment to support data capturing. Many facilities reported being short-staffed—most sentinel laboratories do not have a designated data officer whose primary role is to capture and safeguard data generated in the lab. As a result, the AMR surveillance system often relies heavily on volunteers, many of whom are not compensated for these additional responsibilities. While such volunteering responsibilities may appear beneficial initially, they introduce several challenges for data captured into the LIMS. As the participants remarked, no one can be held responsible for shortcomings or delays in data entry without formal accountability structures. Volunteers also often prioritize their primary duties over AMR data entry, resulting in delays in reporting and a missed opportunity for the lab and the National Coordinating Center to monitor disease outbreaks in real-time. The laboratory's inability to capture quality data also impacts its ability to demonstrate the benefits of AMR surveillance, gain institutional support, and reinforce a cycle of disadvantage.

The lack of institutional support is also evident in the availability of required consumables for routine laboratory testing. The National Coordinating Center at NCDC vets prospective sentinels to ensure that they are at least minimally resourced to contribute to antimicrobial resistance surveillance. However, one participant notes that their facility was not adequately prepared to join the sentinel network, which significantly impacted the timely resourcing of consumables in their laboratory and the ability to culture pathogens for capture in the LIMS. This is partly because *"labs do not have a stock management system in place to keep track of inventory and wholly because getting hold of suppliers and vendors to deliver reagents and consumables is tasking, in Nigeria".* Two out of seven sentinel labs assessed (one was inaccessible for assessment) lacked stock management mechanisms, which disrupted their ability to provide adequate, proper patient care and services. Consequently, when labs run out of essential reagents or consumables for accurate testing and diagnosis, they often resort to using less reliable tests, which results in inaccurate diagnoses and compromises data quality. Additionally, when facilities experience stock-outs, especially with antibiotic disks, they are unable to adequately capture laboratory and AST information, which limits patient care and hinders quality management at facilities and the coordinating center.

### 3.3. Addressing (limited/non-) use of laboratory information management systems

Fig 3 summarizes findings from stakeholder mapping of potential solutions to address the barriers and facilitators of efficient use of the LIMS. For each solution, we reflected on the level of effort required to intervene and the level of influence it would have on behavioral change. We also identified the relevant stakeholders for each solution.

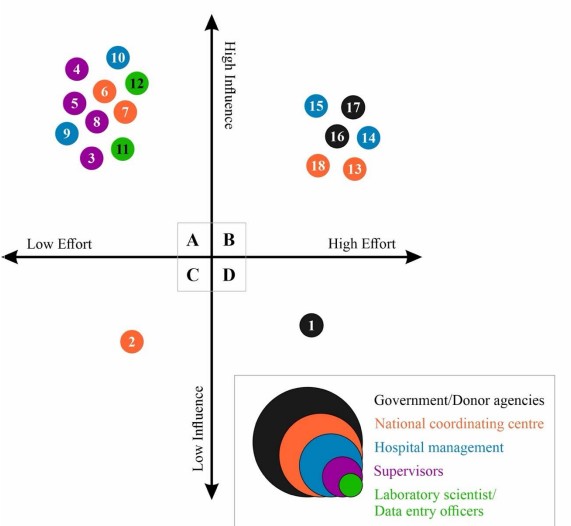

**Fig 3. Stakeholder mapping** *(Note: Panel A = Low effort-high influence interventions, Panel B = High effort-high influence interventions, Panel C = Low effort-low influence interventions, Panel D = High effort-low influence interventions. Interventions are color-coded based on the responsible stakeholder. (1)—Provide funding to support Indigenous technology development; (2)—Create a dedicated platform for networking; (3)—Give due acknowledgement to data entry officers in output arising from the data; (4)—Involve data entry officers in data use and evidence generation; (5)—Monitoring internal quality control; (6)—Sensitization on data use. E.g. Annual reports; (7)—Universal SOP on WHONET; (8)—Training and retraining on WHONET software; (9)—Provision of incentives to data officers; (10)—Provision of adequate computing resources; (11)—Familiarize oneself with the system; (12)—Ensure adequate and routine equipment backups; (13)—Develop a dedicated national data center/platform; (14)—Creation of data bench or a dedicated data entry officer;(15)—Ensure adequate supply of consumables; (16)—Provide adequate service contract for preventive maintenance of donated machines; (17)—Ensure training of local engineers to support machine repairs beyond warranty period; (18)—Improve interlinkages with other LIMS like Health in the Box.).*

Panel A included low-effort and high-influence interventions. Particularly noteworthy was that these interventions can be implemented by stakeholders within the AMR surveillance system; input from harder-to-reach government and donor stakeholders is not perceived as required. Supervisors of laboratory staff are critical in this regard. Stakeholders emphasized the crucial importance of engaging data entry officers in data utilization and evidence generation to cultivate a collective sense of ownership of the LIMS and promote behavioral change. This also includes recognizing and/or acknowledging them in publications arising from the use of data from their sentinel laboratory. This perspective aligns with insights derived from the problem identification phase, which revealed a lack of awareness of how collected data is utilized and its potential to inform public health strategies. The development and dissemination of an annual report by the National Coordinating Center, highlighting key insights derived from facility-level data, could also serve as a critical tool to drive behavioral change. Moreover, IPC teams and supervisors could strategically leverage evidence presented in the annual report to advocate more effectively for institutional support and resources from their hospital leadership. One participant from a facility with a dedicated data officer shared their experience as follows: *"I had to convince the hospital management, and I must thank this programme for their support. As a result of their intervention, I wrote to management, highlighting the importance of generating this data [required data for The National Coordinating Center]. Initially, I requested that a medical lab scientist handle the data due to the technical complexities involved. However, management advised that lab scientists would be best suited for laboratory tasks, and it would be more effective to have a dedicated data specialist responsible for data capture. We put him [the data scientist] through the process, and he has been very efficient with it."* Training and retraining laboratory officers and other key stakeholders could also mitigate concerns about limited technical skills, which affect confidence and attitude towards using LIMS. It is particularly important to adapt this training to the local

context and make it more accessible to support quality improvement for decision-making. Existing training pitched at the entry level, for example, only focuses on installation, laboratory configuration, and data entry, while key modules, such as SaTScan for outbreak detection, are treated as advanced topics. Stakeholders remarked that confusion about what is covered in the various modules demotivates enrolment in the more advanced topics, as they perceive them as too complex.

Panel B includes high-effort and high-influence interventions, primarily involving hospital management, the National Coordinating Center, the government, and international donors. Stakeholders noted that it was crucial for the National Coordinating Center to collaborate with the national development team to establish interlinkages between multiple LIMS, thereby reducing the time spent on duplicating efforts and promoting the efficient use of the LIMS. A well-coordinated supply of necessary consumables for routine laboratory testing would also enable laboratories to conduct the required tests and capture necessary fields into the LIMS. An adequate service contract for preventive maintenance and training of local engineers to support machine repairs could also limit frequent machine breakdowns, a crucial barrier identified by all stakeholders. A centralized national data center also represents a high-effort, yet low-influence, intervention. This platform would consolidate LIMS data across facilities, enabling real-time insights for disease surveillance nationwide. As one participant remarked, *"To date, aggregated data is not available. We have come a long way; by now, we should have resolved the issue of how to aggregate data nationally and make it accessible for people to use. For example, someone somewhere should be able to access data for 2018 in their office. The National Coordinating Center does not necessarily need to send the report back to the sentinels but just make it publicly accessible".* Participants also remarked that such a public platform could motivate facilities to capture and submit data using the relevant system through social influence, whereby as one participant notes *"if I know that by a particular date, the national data will be out and my site will not be published (or my row will be NULL), next time, my site will work harder to make sure the data is complete and to get some credit or recognition. Currently, if I send or don't send, nobody knows, but if I know [through the public repository] that 10 of my colleagues have uploaded their data and I have not, I will feel very bad."* The establishment of data benches or the deployment of dedicated data entry officers at health facilities is another essential strategy echoed by all stakeholders to mitigate staffing issues. Although requiring low effort, this intervention has far-reaching influence. Having dedicated personnel ensures accurate and timely data entry, reduces the workload on laboratory scientists, and improves data quality.

Furthermore, Panel C included low-effort and low-influence interventions involving only the National Coordinating Center. Stakeholders in both groups alluded to the importance of having a central networking channel (e.g., via WhatsApp) to foster a sense of community, knowledge exchange, and collective problem-solving. Stakeholders suggested that such a platform would not only support the timely resolution of emerging technical and operational issues but also promote sustained interest in the use of the system.

Lastly, Panel D included high-effort and low-influence interventions that involve government and donor agencies. Participants remarked that investing in indigenous technology development can potentially address many of the equipment-related challenges identified in the problem phase. Adequate funding would foster innovation in the design of systems tailored to Nigeria's unique healthcare infrastructure and challenges, reduce reliance on foreign systems, and ensure adaptability to evolving local needs and availability for repairs when necessary.

## 4. Discussion

Several studies today have examined experiences, barriers, and facilitators of the efficient use of electronic health information systems among early adopters in diverse settings [30–32]. Among other factors, these studies show that power outages, outdated equipment, and pressure from federal programs to establish parallel information systems [30] are key challenges to technology adoption in healthcare settings. In the area of antimicrobial resistance surveillance, the Mapping Antimicrobial Resistance and Antimicrobial Use Partnership (MAAP) survey revealed that very few laboratories in Africa were engaged in bacteriological testing, and only 13% of these laboratories were using electronic laboratory information systems, despite WHONET being a free tool developed specifically for resource-limited settings [8]. The use of laboratory

information management systems in developing countries is an important yet underexplored topic [9,12,33], and the influence of multiple actors on such use has not received sufficient attention in the literature.

Our study thus presents an essential major step in uncovering these dynamics. We leveraged a participatory action research framework by engaging key stakeholders involved in AMR surveillance data generation and use, such as data analysts, laboratory scientists, infection prevention and control specialists, medical doctors, and representatives from the National Coordinating Centre in Nigeria. In the narratives captured through a series of participatory action research phases, stakeholders described a complex interplay of systemic, political, and structural/user behavioral factors that shape the use behavior of the laboratory information management system for AMR surveillance. Precisely, stakeholders identified computing resources, institutional support, laboratory resourcing, and adequate guidance for system use as crucial barriers and facilitators of the use of an electronic laboratory management system. Several of these issues have also been identified and described in the literature as systemic challenges to AMR surveillance in low- and middle-income countries [9,12,33–37]. A recent study has also demonstrated the feasibility of implementing centralized sequencing with decentralized bioinformatics analysis to enhance bioinformatics capacity among public health professionals in resource-limited settings [38]. Consequently, these factors are likely to be indirectly linked with the adoption of technology for AMR surveillance, as sentinel laboratories may lack the necessary infrastructure to generate the requisite data, which, in turn, undermines their capacity to fully utilize laboratory information management systems.

In addition, the incompatibility of existing laboratory information management systems often leads to significant frustration, resulting in the duplication of efforts and, ultimately, an unsustained use of the LIMS. This finding is consistent with the facilitating condition, compatibility, and perceived behavioral control constructs in the Unified Theory of Acceptance and Use of Technology [18], the Theory of Reasoned Action [15], and the Technology Acceptance Model [16,17], respectively. Moore and Benbast [39] particularly argue that the degree to which an innovation is perceived to be compatible with existing experiences and needs shapes technology use behavior. In a recent review, Vong et al. [12] also found that technical problems (such as the configuration of WHONET and BacLink), system interoperability, lack of data standards, and lack of a well-trained local and national IT workforce to advise and lead the process, limited human resources shape technology adoption for AMR surveillance in South East Asia. Although BacLink [11], an associated tool, provides a linkage between WHONET and other existing LIMS or hospital information management systems, a few other sentinel labs reported using LIMS that are inoperable or believed to be inoperable with WHONET.

Our analysis also showed that the lack of engagement, feedback, and acknowledgement shapes system use. The extent to which the use of an innovation or technology is perceived to enhance one's image or status in their social system and its link to improved system use is well documented in the image construct of the innovation diffusion theory and the social influence construct of the UTAUT model. Do et al. [35] particularly found that adequate involvement and collaboration with the relevant stakeholders improved AMR surveillance. Addressing these challenges is crucial to improving user behavior, supporting the collection of accurate and consistent data, enhancing AMR surveillance, and ultimately improving the ability to detect and respond to AMR-related outbreaks. Incentivizing data officers and laboratory scientists has the potential to motivate employees, highlighting that they are effective and imperative contributors to achieving organizational objectives and purposes. Studies on social recognition and employee engagement have also shown that receiving appreciation signals an employee's uniqueness and individuality and can positively influence an employee's perception of self-worth and identity, promote competitiveness, and enhance trust and productivity [40,41]. More importantly, acknowledging all stakeholders, including data officers and laboratory scientists, in publications resulting from data use can motivate the efficient use of the system, since people desire to be perceived as talented, competent, and intelligent [42].

Greater and effective investment in health systems and services is also essential for implementing key interventions, including staff training to develop skills and use technology, equipping facilities, building organizational capacity, alleviating staff overload through hiring, and upgrading infrastructure. Addressing these challenges would not only enhance data-capturing efficiencies but also support sentinel labs to devote more time to generating novel insights from their facility

data. For example, narratives from our study show that many sentinel labs are unable to use applications within the WHO-NET tool that generate local antibiograms and identify outbreaks. These are considered 'advanced topics', but they are also the most useful capabilities of the tool at the facility level. Without implementing these functions, surveillance data are not actionable locally. Data officers also describe spending hours in WHONET training courses learning how to generate configuration files, which could be made for them and distributed from the National Coordinating Center. Going forward, we propose that WHONET training be redesigned to start with these analyses (using dummy data) and have routine data taught after these processes are mastered so that local use of surveillance data is enabled and the value of data scientists becomes more visible.

Although our study generates new insights into technology adoption for AMR surveillance, several aspects of the present study limit the extent to which our findings can be generalized. Our study focused solely on a subset of early adopters of WHONET in the Nigerian AMR surveillance system and may not accurately represent the diverse issues experienced by other regions or institutions. All the sentinels are located in urban areas, where users are likely to have some experience with technology use. It is likely that we will identify more barriers and facilitators as surveillance expands to remote areas with limited connectivity. Furthermore, there was limited involvement of the National Coordinating Center in in-depth interviews and the participatory co-design workshop. Although their broader system-level perspective may have provided additional insights beyond those captured from frontline stakeholders, we prioritized perspectives from stakeholders who were directly involved in using WHONET to capture laboratory data in these phases. Nevertheless, their participation in the feedback session provides an opportunity to gather their input and reflections on the proposed solution. Lastly, our study sought to identify factors that affect effective and efficient data flow and find collective solutions to reducing or ending these issues. However, these solutions are yet to be tested, and whether they will lead to actual change is unknown.

Nevertheless, we believe that our participatory action research approach to identifying these solutions and the responsible actor is likely to motivate behavior change, especially given that the majority of the interventions identified require low effort. As a result, the co-developed solutions could provide a roadmap for targeted interventions to strengthen AMR surveillance capacity and inform evidence-based public health strategies as global efforts to improve laboratory and surveillance capacities expand across LMICs. Similarly, our study highlights the crucial role of the participatory action research approach in generating evidence. By involving diverse stakeholders in co-identifying key barriers and facilitators to the use of LIMS, our work also helps them to understand how they form part of a larger system and their capacity to effect change within their respective systems.

## Conclusion

Our findings contribute to a rich body of literature on barriers and facilitators to digital technology application for antimicrobial resistance surveillance, particularly in a resource-limited setting. By illuminating how a multitude of systemic, political, and structural factors intersect with user-related factors across various stakeholders, we demonstrate the importance of a participatory action research approach that incorporates the voices of multiple stakeholders. Stakeholders also identified 18 potential solutions to tackle key challenges, ten of which require low effort and have a high influence on LIMS use behaviors. These solutions provide a roadmap for targeted interventions to strengthen AMR surveillance capacity and inform evidence-based public health strategies.

## Supporting information

**S1 Text. Key informant interview guide.**
(DOCX)

**S2 Text. PAR workshop discussion guide.**
(DOCX)

## Acknowledgments

We thank Elizabeth T Akande, Funmi Adekunle, Jola-Ade J Ajiboye, Olajumoke L. Oladele, and Mariam A. Odebode for their technical and logistical assistance. We also acknowledge support from the staff and management of the facilities where this study was conducted for their valuable contributions and insights.

## Author contributions

**Conceptualization:** Iruka N. Okeke.

**Data curation:** Ayodele Akinyele, Abeeb A. Adeniyi, Oluwatobiloba S. Kazeem, Michael Popoola, Tochukwu C. Agboeze.

**Formal analysis:** Ayodele Akinyele, Emmanuel Olamijuwon.

**Funding acquisition:** Iruka N. Okeke.

**Methodology:** Ayodele Akinyele, Emmanuel Olamijuwon.

**Project administration:** Ayodele Akinyele.

**Resources:** Iruka N. Okeke.

**Supervision:** Iruka N. Okeke.

**Validation:** Oluwatobiloba S Kazeem, Michael Popoola.

**Visualization:** Emmanuel Olamijuwon.

**Writing – original draft:** Ayodele Akinyele, Emmanuel Olamijuwon, Abeeb A. Adeniyi.

**Writing – review & editing:** Emmanuel Olamijuwon, Iruka N. Okeke.

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
