## [Decision Letter · Decision Letter 0]

PGPH-D-25-00848

Barriers and facilitators to digital technology application for antimicrobial resistance surveillance: a co-produced qualitative synthesis

Dear Dr. Okeke,

Thank you for submitting your manuscript to PLOS Global Public Health. After careful consideration, we feel that it has merit but does not fully meet PLOS Global Public Health’s publication criteria as it currently stands. Therefore, we invite you to submit a revised version of the manuscript that addresses the points raised during the review process.

We look forward to receiving your revised manuscript.

Kind regards,

Sadia Shakoor

Academic Editor

Journal Requirements:

Additional Editor Comments (if provided):

Reviewers' comments:

Reviewer's Responses to Questions

**Comments to the Author**

1. Does this manuscript meet PLOS Global Public Health’s publication criteria ? Is the manuscript technically sound, and do the data support the conclusions? The manuscript must describe methodologically and ethically rigorous research with conclusions that are appropriately drawn based on the data presented.

Reviewer #1: Yes

Reviewer #2: Yes

2. Has the statistical analysis been performed appropriately and rigorously?

Reviewer #1: Yes

Reviewer #2: Yes

3. Have the authors made all data underlying the findings in their manuscript fully available (please refer to the Data Availability Statement at the start of the manuscript PDF file)?

Reviewer #1: Yes

Reviewer #2: Yes

4. Is the manuscript presented in an intelligible fashion and written in standard English?

Reviewer #1: Yes

Reviewer #2: Yes

5. Review Comments to the Author

Reviewer #1: The manuscript by Ayodele Akinyele et al. describes the barriers and facilitators to the application of digital technology, particularly WHONET, for antimicrobial resistance (AMR) surveillance. The authors conducted a well-designed social science study, and the manuscript is very well written. It will be important to the field. The introduction, in particular, is exceptionally well written. I have a few major and minor comments for consideration.

Major comments:

[1] Baseline demographics of the nine participating sentinel sites and 23 participants in the in-depth interviews are largely unclear. Table 1 notes that 19 participants were "biomedical scientists," while the text on line 7 of page 9 refers to them as "laboratory scientists." Are they different?

Additionally, it is stated that "most" perform data capture "voluntarily." It is unclear to readers what the "full-time jobs" of these 23 participants (and 4 data officers) are? Who pays their salaries? Universities as academic staff or Hospitals as laboratory staff (but called scientists / data officers)? How many of them are full-time academic staff whose roles include research, teaching, and publishing manuscripts? How many are full-time laboratory staff hired by university hospitals to perform routine laboratory work, data entry, data analysis, and report submissions? Do the university hospitals or university laboratories hire any non-academic persons to do such tasks? Are any of the full-time academic staff funded by AMR grants or the Fleming Fund, with FTE dedicated to supporting WHO GLASS per contract, to do such tasks? Can this part be clarified?

[2] Although the participants in the in-depth interviews are 'voluntary researchers' and noted that their time supporting 'data capture' is voluntary, it is possible that these nine participating sentinel sites have full-time laboratory personnel tasked with 'data entry,' 'data analysis,' and 'report submission.' Can the availability or lack of full-time laboratory personnel be clarified with proper numerators and denominators?

For example, "XXX out of nine sentinel laboratories have at least one full-time laboratory personnel responsible for 'data analysis' and 'report submission' to the National Coordinating Centre, while the remaining XXX sentinel laboratories do not."

Although this is a social science study, baseline information is important for readers to understand the context. This is exemplified on page 12, line 20: "Two out of seven labs assessed did not have stock management mechanisms in place." More descriptions using denominators of all labs with data would be helpful to readers and provide more impact to policymakers. (I assume that the remaining two labs do not have data about stock management mechanisms, if not typo. Can this point be clarified in the text?)

[3] As the key topic of the study is WHONET and LIMS, can the authors describe clearly from the interviews

What LIMS (name of the LIMS and companies) are being used daily in the nine study labs?

Do any labs use MS Excel or MS Access as LIMS?

Do all nine labs perform double data entry in WHONET and their own LIMS?

Do none of the nine labs use BacLink? Although page 10, lines 15-26, describes some issues about LIMS, Health-in-a-Box, and BacLink, and it is clear that all hospitals use "Health-in-a-Box" as a Hospital Management Information System, it is unclear how many are using LIMS, which LIMS, and BacLink. BacLink could be difficult, but "many" (line 23) is unclear whether none can use it at all in Nigeria.

Are there any hospitals performing triple or quadruple data entry (e.g., Health-in-a-Box, LIMS, WHONET, and notebooks)?

Can this part be clarified with proper numerators and denominators (by number of hospitals)?

[4] I am uncertain about the question, "Have the authors made all data underlying the findings in their manuscript fully available?" Although the authors stated that "All data used for the study are included in the paper," interview guides (including question list and other related documents), individual-level anonymous data/quotes from participants are not provided or deposited in a public repository. The data and quotes related to each theme are not fully available; only few selected quotes are shown in the text. I wonder whether the authors could provide more supporting data as an appendix or to a public repository. This is highlighted by the fact that readers could be unclear whether other points were asked in the interview and what the anonymous answers/quotes to those points were.

Minor comments:

[1] The terms LIMS and LIS (page 15, lines 2 and 4): Is LIS a typo? Do the authors mean LIMS? If the authors intend to differentiate between LIS and LIMS, can they provide definitions or details about the differences between LIS and LIMS?

[2] Can the authors clarify in intro methods and/or discussions that LIMS in this work focuses solely on "microbiology LIMS," which can differ from general LIMS used in clinical laboratories? Did this distinction cause confusion for any participants? Do the laboratories in Nigeria use different general LIMS and microbiology LIMS?

If yes, can this be added into the discussion?

[3] Can the authors describe or discuss whether Health-in-a-Box includes functions for both "general LIMS" and "microbiology LIMS"? Can details obtained from participants during the interviews be provided to readers in the manuscript?

[4] The authors frequently use the term "capture data" (e.g., page 9, line 10). Can the authors clarify whether this means "input data," "export data," or "data analysis"? For example, do the interviewees (academic scientists) support the hospital laboratories by inputting data into WHONET, exporting data from LIMS, importing data into WHONET, and analyzing data using WHONET? Or some functions of those? Or do they mean something else by "capturing data"?

[5] Acronym of LIMS (page 16, lines 27 and 28): Should "laboratory information management systems" be abbreviated as LIMS here?

Reviewer #2: The participation from the National coordinating center is 0 at the interview and participatory co-design workshop stage though their participation in the feedback session is clear. This could be a limitation that prevented their perspective from being included. this must be mentioned as a limitation.

6. PLOS authors have the option to publish the peer review history of their article (what does this mean? ). If published, this will include your full peer review and any attached files.

**Do you want your identity to be public for this peer review?** For information about this choice, including consent withdrawal, please see our Privacy Policy .

Reviewer #1: No

Reviewer #2: No

---

## [Editor Report · Decision Letter 1]

Barriers and facilitators to digital technology application for antimicrobial resistance surveillance: a co-produced qualitative synthesis

PGPH-D-25-00848R1

Dear Prof Okeke,

We are pleased to inform you that your manuscript 'Barriers and facilitators to digital technology application for antimicrobial resistance surveillance: a co-produced qualitative synthesis' has been provisionally accepted for publication in PLOS Global Public Health.

Best regards,

Sadia Shakoor

Academic Editor